# Prenatal and Postnatal Hair Steroid Levels Predict Post-Partum Depression 12 Weeks after Delivery

**DOI:** 10.3390/jcm8091290

**Published:** 2019-08-23

**Authors:** Leila Jahangard, Thorsten Mikoteit, Saman Bahiraei, Mehrangiz Zamanibonab, Mohammad Haghighi, Dena Sadeghi Bahmani, Serge Brand

**Affiliations:** 1Research Center for Behavioral Disorders and Substance Abuse, Hamadan University of Medical Sciences, 6516848741 Hamadan, Iran; 2Center for Affective, Stress and Sleep Disorders, Psychiatric Clinics, University of Basel, 4502 Basel, Switzerland; 3Psychiatric Hospital Solothurn, University of Basel, 4503 Solothurn, Switzerland; 4Department of Obstetrics & Gynecology, Hamadan University of Medical Sciences, 6516848741 Hamadan, Iran; 5Health Institute, Substance Abuse Prevention Research Center, Department of Psychiatry, Kermanshah University of Medical Sciences (KUMS), 6719851115 Kermanshah, Iran; 6Sleep Disorders Research Center, Department of Psychiatry, Kermanshah University of Medical Sciences (KUMS), 6719851115 Kermanshah, Iran; 7Alzahra Research Institute, Neurosciences Research Center, Isfahan University of Medical Sciences, 8174675731 Isfahan, Iran; 8Division of Sport Science and Psychosocial Health, Department of Sport, Exercise and Health, University of Basel, 4052 Basel, Switzerland

**Keywords:** pre- and postnatal hair steroids, post-partum depression, cortisol, cortisone, testosterone, progesterone, DHEA

## Abstract

Background: Within three to six months after delivery, 13%–19% of women suffer from post-partum depression (PPD), understood as a dysfunctional adaptation to the postpartum condition and motherhood. In the present cross-sectional study, we compared the hair steroid levels of women 12 weeks before and after delivery and with or without PPD. Method: The present study was a cross-sectional study conducted twelve weeks after delivery. At that time, 48 women (mean age: 25.9 years) with PPD and 50 healthy controls (mean age: 25.2 years) completed questionnaires on depressive symptoms. Further, at the same time point, 6 cm lengths of hair strands were taken, providing samples of hair steroids 12 weeks before and 12 weeks after delivery in order to analyze hair steroids (cortisol, cortisone, progesterone, testosterone, and dehydroepiandrosterone (DHEA)). Results: Compared to those of women without PPD, hair steroid levels (cortisol, cortisone, progesterone) were significantly lower in women with PPD both before and after delivery. Lower prenatal cortisone and progesterone levels predicted higher depression scores 12 weeks after delivery. Lower prenatal levels of cortisol and progesterone and higher levels of DHEA, and postnatal lower levels of cortisol, cortisone, and progesterone, along with higher levels of DHEA predicted PPD-status with an accuracy of 98%. Conclusions: PPD is associated with blunted hair cortisol, cortisone, and progesterone secretions both pre- and postpartum. Such blunted steroid levels appear to reflect a stress responsivity that is less adaptive to acute and transient stressors. It follows that prenatally assessed low hair cortisol and progesterone levels, along with high DHEA levels, are reliable biomarkers of post-partum depression 12 weeks after delivery.

## 1. Introduction

It is estimated that a few days after delivery 40%–80% of mothers suffer from symptoms of depression (“baby blues”). Furthermore, 13%–19% suffer from post-partum depression (PPD) in the first six months following delivery [1,2,3]. Typically, women with PPD exhibit fewer positive emotions and more negative emotions towards their child, and they are less responsive and sensitive to the newborn’s behavior and interactional efforts [1,2,3]. Additionally, mothers with PPD display a sad and depressed mood, a virtually complete loss of activity, fatigue, and loss of energy, feelings of worthlessness and guilt, poor cognitive performance, severe sleep disturbance, and signs of suicidal behavior [4]. Heron et al. [5] have suggested that symptoms of depression and anxiety before pregnancy could predict the emergence of PPD. From the perspective of evolutionary psychology and psychiatry [6,7,8,9] and bearing in mind humans’ ancestral environment, PPD could be seen as an adaptive response: Withdrawing from social activities as a result of real or perceived lack of social support and reducing ‘investment’ in the newborn both reduces social stress and prompts the mother’s community to invest more effort and resources both in the newborn and the mother [6]. In contrast, from a psychiatric point of view, PPD is regarded as a major depressive disorder, with the specifier of onset four weeks after delivery [4], and, as such, must be treated following evidence-based algorithms. To further support such algorithms, research on psychophysiological variables has been undertaken to explain the emergence and maintenance of PPD [10,11,12].

Specifically, neuroendocrine changes are attracting growing interest in PPD research [13]. Research on the role of neuroendocrines (and particularly cortisol) in major depressive disorders [14] and post-traumatic stress disorder [15] has indicated that an altered hypothalamus-pituitary-adrenocortical axis activity (HPA AA) is associated with these affective disorders [16]. In the present study, we focused on steroids (cortisol, cortisone, progesterone, testosterone and dehydroepiandrosterone or DHEA), and we examined the relation of levels of these steroids with symptoms of depression (both antepartum and postpartum). To this end, we analyzed steroids in hair strands, as hair strands are considered a retrospective calendar and thus provide insight into changes in steroid secretion over time and as a function of women’s long-term affective states. [10,17,18]. Furthermore, steroid concentrations in hair are robust against short-term psychophysiological fluctuations, and hair strands are easy to collect, transport, and preserve at room temperature [10,17,18]. In contrast, steroids in blood, urine or saliva might reflect current physiological and psychological states, circadian rhythms of the time of the day of sampling, and situational circumstances, such as food intake and subjectively perceived stress [19,20].

In the following sections, we provide a brief overview of studies on steroids during and after pregnancy. Such steroids, and above all, cortisol, were mainly assessed in blood, saliva, and urine, and, to a much lesser extent, in hair.

### 1.1. Plasma Cortisol in Healthy Pregnant Women

Focusing on *plasma cortisol* in healthy pregnant women, Jung et al. [21] found an increase in free cortisol and corticosteroid-binding globulin (CBG) from the second to the third trimester. These authors also briefly summarized previous publications and concluded that this rise in total plasma cortisol in healthy pregnant women appears to be related to up-regulation of the maternal hypothalamic-pituitary-adrenal axis in addition to elevated CBG. Furthermore, cortisol levels in the last trimester may be as much as two to three times higher than in the first trimester. Specifically, the up-regulation of HPA AA leads to an increase in total plasma cortisol. However, the parallel rise in CBG levels buffers these increasing levels of cortisol. As a result, the increase in active and free cortisol is less pronounced. This mechanism protects both mother and infant against detrimental exposure to high cortisol levels. Thus, increases in plasma cortisol from the second to the third trimester reflect physiological adaptations to pregnancy among healthy women.

### 1.2. Salivary, Blood and Urine Cortisol in Pregnant Women with Major Depressive Disorders

Complementary to this, Tsubouchi et al. [11] observed lower antenatal salivary cortisol levels in women suffering from major depressive disorders during the second and third trimester. Likewise, Glynn et al. [22] found lower saliva cortisol levels in women with PPD, consistent with the notion of hypocortisolism in such women. Hypocortisolism in women with PPD appears to be neuro-endocrinologically closer to post-traumatic stress disorder (PTSD) than to MDD [15,16,23]. Thus, we expected in the present study to find that participants with PPD would have lower hair cortisol levels during their third trimester than would healthy mothers.

Focusing further on salivary and blood cortisol and PPD, Seth et al. [13] conducted a systematic review of 47 studies focusing on cortisol levels and symptoms of PPD. Seth et al. [13] found that methodological differences (sample size, assessment of symptoms of depression, methods of cortisol sampling (saliva, blood), and assessment time after delivery) made it difficult to identify any clear-cut pattern of results. Nevertheless, they concluded that hypercortisolemia appears to be associated with transient depressive states, while the hypocortisolemia appears to be related to chronic PPD.

In contrast, in a systematic review of possible biomarkers of post-partum depression, Serati et al. [24] found that the pattern of associations between HPA AA (with saliva and blood cortisol levels as outcome variables) and PPD was unclear. However, Szpunar and Parry [25] concluded in their systematic review that there was converging evidence of lower morning blood cortisol levels in women with PPD than in healthy controls, while for other hormones, such as prolactin and thyroid hormones, the pattern of results was less clear. Specifically, Szpunar and Parry [25] commented on two studies by Field et al. [26,27], in which women with PPD had higher urinary cortisol levels during the second, but not during the third, trimester, when compared to healthy controls, thus pointing to a down-regulation of HPA AA 12 weeks before delivery. However, Field and colleagues were unable to replicate these findings in a later study [28], indicating that contradictory findings can arise even within the same research group.

Duthie and Reynolds [29] concluded from their systematic review that the experience of chronic stressful life events during early pregnancy is associated with a blunted peaking of salivary cortisol levels in the morning. The experience of positive life events reduces morning salivary cortisol levels, while the experience of negative life events is not associated with morning salivary cortisol levels. Duthie and Reynolds [29] concluded that different stressors appear to have differing effects on the HPA AA among women during and after pregnancy.

Collectively, the pattern of results from cortisol concentrations in saliva, blood, and urine in pregnant women with PPD does not seem to be fully consistent. Nevertheless, a hypercortisolemia was mainly associated with transient symptoms of depression, while a hypocortisolemia was associated with chronic symptoms of depression.

### 1.3. Hair Cortisol in Women with Post-Partum Depression

Research on hair steroids, and specifically cortisol, in women with PPD is inconclusive and scarce, compared to studies on the HPA AA in women with PPD focusing on saliva, urine, and blood cortisol [30,31]. This is surprising since hair cortisol concentrations provide a unique objective biomarker for the analysis of endogenous cortisol levels for clinical diagnostics and research [32]. Although Stalder et al. [19] cautioned that sex, age, hair washing frequency, hair treatment, and contraceptive use could all modify hair cortisol levels, Braig et al. [18,33] claimed that hair cortisol concentrations are robust across a broad range of confounding influences, such as gender, hair color, age, or hair treatment.

Caparros-Gonzalez et al. [34] assessed 44 women postpartum at either low or high risk of post-partum depression. Hair strands were taken 16 days after delivery; these hair strands reflected the hair cortisol concentrations of the three trimesters of pregnancy. Caparros-Gonzalez et al. [34] observed that hair cortisol concentrations increased continuously in women without symptoms of PPD from the first to the third trimester. This pattern of increasing cortisol concentrations during pregnancy among healthy women was further confirmed by Duthie and Reynolds [29]. In contrast, Caparros-Gonzalez et al. [34] showed that in women with symptoms of PPD, hair cortisol concentrations were consistently and significantly higher than the concentrations for women without PPD. Furthermore, a multiple regression analysis revealed that higher hair cortisol concentrations during the first trimester predicted the risk of suffering from symptoms of PPD. An additional result was that higher hair cortisol concentrations were associated with stronger symptoms of subjective pregnancy-specific stress and stronger symptoms of depression and anxiety. Likewise, Kalra et al. [35] observed that, in a sample of 25 healthy women, higher hair cortisol concentrations during pregnancy were associated with higher perceived stress.

In contrast to the results of Caparros-Gonzalez et al. [34], Braig et al. [33] were unable to find any kind of association between hair cortisol concentrations and self-reported symptoms of chronic stress, anxiety, or depression in a sample of 768 participants. To explain these null findings, Braig et al. [33] argued that the completion of self-rating questionnaires about 1.5 days after delivery might have biased recall from long-term memory.

Braig et al. [33] also reviewed previous studies and concluded that symptoms of anxiety and depression might be associated with both a hypo- and hyperactivity of the HPA AA, as measured by hair cortisol concentrations.

Orta et al. [36] assessed hair cortisol concentrations in 97 pregnant women, separately for all three trimesters, and observed that higher scores for general anxiety and perceived stress were associated with lower hair cortisol concentrations. Again, these results are a good match with findings reported elsewhere [19,37]. Lower hair cortisol concentrations appeared to be associated with chronic stress [38,39,40] and psychiatric disorders, such as generalized anxiety disorders [41,42] and PTSD [43,44,45,46].

To summarize, studies on hair cortisol during and after pregnancy are scarce and the pattern of results appears inconsistent. Such inconsistent findings appear to reflect, above all, methodological issues, such as the timing of psychological assessments and the lack of a thorough diagnosis of PPD.

### 1.4. Other Steroids during Pregnancy and after Delivery

For other steroids, such as cortisone, progesterone, testosterone, and dehydroepiandrosterone (DHEA), research with healthy participants has indicated increased levels during the second and third trimester and a drop hours or days after delivery [21,22,47]. However, all these studies were based on steroids in blood or saliva, not in hair, and the pattern of results remained complex and unresolved [48]. Furthermore, Schiller et al. argued that such a heterogenous pattern of results may also be due largely to the complex interplay of changes in reproductive hormones during pregnancy and after delivery, along with the complex impact of such hormones and steroids on the entire biological system associated with the occurrence (or lack) of PPD. For example, progesterone is known to regulate the synthesis, release, and transport of neurotransmitters associated with mood regulation [49] and rapid changes in progesterone concentrations are associated with an increased risk of PPD [48]. Next, concentrations of the neurosteroid dehydroepiandrosterone (DHEA) were significantly lower in individuals with MDD, while the association between DHEA concentrations and the occurrence of PPD has yet to be determined [48].

It also appears that testosterone concentrations have not been a focus in the search for possible biomarkers to explain the occurrence of PPD.

Next, Guintivano et al. [50] were unable to identify any differences in the plasma concentrations of progesterone and ALLO (allogregnanolone; a neurosteroid similar to DHEA) in women with and without PPD six weeks after delivery.

Importantly, from previous research, it has emerged that such steroids have been assessed at different time points, with different criteria to diagnose PPD, and different methods of sampling (blood, saliva; but not hair), making it virtually impossible to compare results across studies.

To conclude, research on further steroids, such a cortisone, progesterone, testosterone, or DHEA, is scarce, and results are inconsistent and based on blood and saliva samples, while data from hair samples are currently missing.

### 1.5. Hypotheses of the Current Study

The following four hypotheses and two research questions were formulated. First, following Glynn et al. [22] and Orta et al. [36], we expected that, compared to participants without PPD, participants with PPD would show lower hair cortisol concentrations 12 weeks before delivery. Second, following Seth et al. [13] and Serati et al. [24] we also expected that, compared to participants without PPD, participants with PPD would also show lower hair cortisol levels 12 weeks after delivery. Third, based on previous studies [13,21,24], we expected that levels of cortisol would remain low and unchanged in participants with PPD from the pre- to the post-partum stage, while in participants without PPD, we expected a drop in high cortisol concentrations at prenatal stage to low cortisol concentrations at the postnatal stage [29,34]. Fourth, following others [35,51,52,53] we anticipated that, compared to participants without PPD, participants with PPD would also report they had symptoms of depression before pregnancy. We explored the question of whether levels of other steroids (cortisone, testosterone, progesterone, or DHEA) either change from the pre- to postnatal stage or whether levels of other steroids differ between participants with and without PPD. The second exploratory question was whether any prenatal or postnatal steroid levels could predict scores on the Edinburgh postnatal depression scale (EPDS, continuous dimension) and PPD status (yes vs. no; categorical variable).

## 2. Methods

### 2.1. Procedure

Twelve weeks after delivery, women giving birth at the Department of Obstetrics and Gynecology of the Hamadan University of Medical Sciences (Hamadan, Iran) were approached between January 2017 and January 2019 to participate in the present cross-sectional study on the associations between hair steroid levels and post-partum depression. Women eligible to participate were fully informed about the aims of the study and the confidential and secure data handling. Thereafter, participants signed a written informed consent. Experienced psychiatrists and clinical psychologists conducted a psychiatric interview to identify participants with major depressive disorders and the specifier of onset within four weeks after delivery and to ascertain the diagnosis of PPD, using the Mini International Neuropsychiatric Interview [54]. Participants then completed a series of questionnaires covering sociodemographic information, and symptoms of depression (see details below). Next, hair strands were taken from the occipital region of the scalp (see details below) and assessed for hair steroid levels. The ethics committee of the Hamadan University of Medical Sciences approved the study (HUMS; Hamadan, Iran; IR.UMSHA.REC.1395.578), which was performed in accordance with the principles laid down in the seventh and current edition (2013) of the Declaration of Helsinki.

To establish the quality of the study, we completed the checklist of the Systematic Assessment of Quality in Observational Research (SAQOR; [55], achieving a score of 16 out of a maximum possible 17 (see Appendix A).

### 2.2. Sample

A total of 495 women were approached 12 weeks after delivery. Inclusion criteria for participants with PPD were as follows: 1. Age between 18 and 35 years; 2. vaginal delivery occurred 12 weeks before; 3. willing and able to comply with the study conditions, specifically, to provide hair strands; 4. had a healthy newborn at the moment of delivery, as ascertained by an Apgar-score of eight or higher (see Table 1); 5. minimum gestational age: 37 weeks; 6. diagnosis of post-partum depression, as ascertained by a psychiatrist or clinical psychologist; 7. Edinburgh Postnatal Depression scale (EPDS; see below) score of 12 points or higher; 8. Beck Depression Inventory (BDI; see below) score of 20 points or higher; 9. signed written informed consent. Exclusion criteria were: 1. Acute suicidality; 2. comorbid psychiatric issues, such as bipolar disorder, substance use disorder, or acute psychosis.

Inclusion criteria for participants without PPD were as follows: 1 to 5 and 9 as for the PPD group; 6. Edinburgh Postnatal Depression scale (EPDS; see below) score of 5 points or lower; 7. Beck Depression Inventory (BDI; see below) score of 10 points or lower. Exclusion criteria were the same as for the PPD group.

Being primipara or multipara was not an inclusion or exclusion criterion.

Of the 495 women approached, 65 fulfilled the inclusion criteria for post-partum depression (13.3% of the approached sample); of these, 48 agreed to participate in the study (73.48% of those fulfilling the criteria of PPD), and 17 refused to participate (26.52%). Additionally, of the 495 women approached, 41 (8.3%) did neither fulfill the criteria for participants with or without PPD, and 389 fulfilled the criteria for participants without PPD (78.43% of the approached sample); 112 refused to participate (28.8%), and 50 participants were randomly selected from the remaining pool of 277 possible participants without PPD. Randomization occurred based on the following criteria: matching the group of participants with PPD as regards age, gestational age, and Apgar-score 10’ after delivery. A total of 167 participants fulfilled these criteria, and their codes were put in sealed and separate envelopes; sealed envelopes were put in an opaque ballot box and stirred. A staff member of the hospital not otherwise involved in the study picked out 50 envelopes.

### 2.3. Sample Size Calculation

Sample size calculation was performed with G*Power [56]. We anticipated a large effect size for EPDS scores of *d* = 0.8 [57], an alpha of 0.05, and a Power (1-beta error probability) of 0.95. This yielded a total sample size of 84. To counterbalance possible dropouts and withdrawals, the sample sizes were set at 50 participants with PPD and 50 without PPD.

### 2.4. Tools

#### Self-Ratings of Symptoms of Depression

##### Edinburgh Postnatal Depression Scale (EPDS)

Participants completed the EPDS [58], which consists of ten items. The anchor points for each item are 0 and 3. Higher total scores reflect more marked symptoms of postnatal depression. The maximum score is 30, and a score of 10 or more is taken to indicate the presence of postnatal depression. However, we employed the more restrictive cut-off score of 12 for the self-rating of post-partum depression; others recommend to use a cut-off score of 13 [59]. For the self-identification of no PPD, we applied a cut-off score of 5 (Cronbach’s alpha = 0.84).

##### Beck Depression Inventory (BDI)

Patients completed the BDI [60] (Farsi version; for psychometric properties of this version see Ghassemzadeh et al. [61]). The questionnaire consists of 21 items and assesses different dimensions of depressive symptomatology such as depressed mood, loss of appetite, sleep disorders, and suicidality. Each question has a set of at least four possible responses varying in intensity; e.g., “sadness”: 0 = “I do not feel sad”; 1 = “I feel sad”; 2 = “I am sad all the time and I can’t snap out of it”; 3 = “I am so sad or unhappy that I can’t stand it”. Higher scores reflect greater severity of depressive symptoms (Cronbach’s alpha = 0.90).

##### Symptoms of Depression in the Past

Participants were asked if, in the past (that is before pregnancy), they had been diagnosed with a major depressive disorder or if they had undergone psychological or pharmacological interventions to treat symptoms of depression. Answer was ‘yes’ or ‘no’.

### 2.5. Apgar Score

The Apgar score is a quick method of summarizing a newborn’s health. The acronym stands for A(ppearance), P(ulse), G(rimace), A(ctivity), R(espiration). The sum score ranges from 0 to 10. Scores of 7 and higher are generally normal, scores between 4 and 6 are fairly low, and scores of 3 and lower are critically low (psychometric properties of the Farsi/Persian version: Eslami and Falah [62]. 

### 2.6. Hair Strands Sampling

To sample hair strands, we followed the recommendations of Kirschbaum et al. [63] and others [64,65,66,67]. The steps are as follows:

First, trained staff members divide a participant’s hair at the back of the head using a hair grip.

Second, two to three hair strands are separated close to the participants’ scalp in the posterior vertex region. Hair strands should be at least 3 mm in diameter; as a rule of thumb, 3 mm is equal to half of the diameter of a pencil.

Third, hair strands are combed.

Fourth, combed hair strands are bundled with a prepared packthread loop.

Fifth, 6 cm of strands are cut as close as possible to the scalp.

Sixth, strands are placed in prepared aluminum foil.

Seventh, the scalp near end with the packthread loop is clearly marked with a water-proof marker.

Eighth, the aluminum foil is folded and both ends are sealed.

Ninths, all aluminum foils are consecutively numbered and labeled.

The 6 cm of hair closest to the scalp was tested for steroid concentrations and provided a value reflecting hair steroid concentrations for the last 2 × 12 weeks, that is to say, 12 weeks before and 12 weeks after delivery (assuming an average growth rate of 1cm/month). All participants had long hair and could provide sufficient material for analysis. Note that 3 cm hair segments close to the scalp correspond to a more recent period in time (postpartum), while 3 cm segments distant from the scalp correspond to a more distant period (antepartum). Consequently, hair strands were cut in 2 × 3 cm, corresponding to the two time periods. The two 3 cm hair strands were separately analyzed.

### 2.7. Assessment of Hair Steroids

Hair steroids were assessed by the biochemical Laboratory of the University of Dresden (Germany) using mass spectrometry (LCMS/MS), as thoroughly described in Gao et al. [67] and elsewhere [64]. Samples were washed in 2.5 mL isopropanol for 3 min, and steroid hormones were extracted from 7.5 mg of whole, non-pulverized hair using 1.8 mL methanol in the presence of 50 µL cortisol-d4, cortisone-d7, testosterone-d5, DHEA-d4, and progesterone-d9 as internal standards for 18 h at room temperature. Samples were spun in a bench top centrifuge (Mikro 22R; Hettich GmbH and Co. KG, Tuttlingen, Germany) at 15.200 g relative centrifugal force for 2 min, and 1 mL of the clear supernatant was transferred into a new 2 mL tube. The alcohol was evaporated at 50 °C under a constant stream of nitrogen and reconstituted with 225 µL double-distilled water, 50 µL of which were injected into a Shimadzu HPLC-tandem mass spectrometry system (Shimadzu, Canby, OR, USA) coupled to an AB Sciex API 5000 Turbo-ion-spray triple quadrupole tandem mass spectrometer (AB Sciex, Foster City, CA, USA) with purification by online solid-phase extraction. The lower limits of quantification (LOQ) of this analysis were below 0.1 pg/mg for cortisol and cortisone; for testosterone, DHEA and progesterone the LOQ were below or equal to 1 pg/mL. The inter- and intraassay coefficients of variance were below 10%, for progesterone, DHEA, and for testosterone the CV% were below 15%. We refer to the publication of Gao et al. [67] for those readers wanting more specific chemical and pharmacological details of the analyses.

### 2.8. Statistical Analysis

To detect possible confounders, preliminary computations were performed. Specifically, a series of Pearson’s correlations was computed between maternal age, gestational age, Apgar scores, symptoms depression, and PPD, and hair steroids. All correlation coefficients were < 0.12, (*p*s > 0.30); accordingly, maternal age, gestational age, and Apgar scores were not introduced as possible confounders.

Next, to compare sociodemographic and depression-related dimensions between participants with and without PPD, a series of *t*-tests was performed with Group (with and without PPD) as factor and age, gestational age, symptoms of depression (Beck Depression Inventory; Edinburgh Postnatal Depression Scale), and infants’ Apgar scores as independent dimensions.

In the next step, we examined whether hair steroid concentrations changed over time and between participants with or without PPD. To this end, a series of MANOVAs for repeated measures was computed with the factors Group (with vs. without PPD), Time (pre- vs. post-natal), and the Group by Time-interactions, and hair steroids (cortisol, cortisone, testosterone, progesterone, and DHEA) as dependent variables, always controlling for baseline values, as these values did descriptively differ between the two groups. For *F*-tests, effect sizes were reported with partial eta-squared (*η*_p_^2^), with *η*_p_^2^ ≤ 0.019, indicating trivial effect sizes, 0.020 ≤ *η*_p_^2^ ≤ 0.059 indicating small, 0.06 ≤ *η*_p_^2^ ≤ 0.139 indicating medium, and 0.14 ≤
*η*_p_^2^, indicating large effect sizes [68]. For *t*-tests, effect sizes were reported with Cohen’s *d*, where effect sizes were evaluated as trivial (*d*s: 0–0.19), small (*d*s: 0.20–0.49), medium (*d*s: 0.50–0.79), or large (*d*s: 0.80 and greater) [68].

Pearson’s correlations were computed for associations between symptoms of depression and hair steroids.

A Chi-square test was computed to determine whether participants with PPD were more likely than participants without PPD to report issues of depression before pregnancy. We ran an odds-ratio to calculate the risk to be assigned to the group of participants with PPD, if issues of depression were also present before pregnancy.

In a further step, to predict post-partum depression scores (Edinburgh Postnatal Depression Scale) as a function of prenatal hair steroid levels, a multiple regression analysis was performed.

In a last step, to predict the PPD status (yes vs. no) as a function of pre- and post-natal hair steroid levels, a binary logistic regression analysis was performed.

The nominal level of significance was set at alpha < 0.05. All statistical computations were performed with SPSS^®^ 25.0 (IBM Corporation, Armonk, NY, USA) for Apple^®^ Mac^®^.

## 3. Results

Generally, descriptive and inferential statistical indices are reported in tables and not repeated in the body of the text.

### 3.1. Sample Characteristics

Table 2 reports the sample characteristics of participants with (*n* = 48) and without (*n*= 50) post-partum depression. A total of 90 out of 98 were primiparae. Eight participants had given birth for a second time.

Participants with and without PPD did not differ with respect to age, gestational age, or infants’ Apgar scores 10 min following delivery. Compared to participants without PPD, participants with PPD had higher depression scores (BDI; EPDS), and immediately after delivery their infants had a lower Apgar score. Additionally, participants with PPD significantly more often had a history of a previous depressive episode.

### 3.2. Hair Steroids, Separately for Participants with (n = 48) and without (n = 50) Post-Partum Depression, and Separately for Prenatal and Postnatal Hair Segments

Table 1 provides the descriptive and inferential statistical overview of hair steroid values, both prenatal and postnatal, and separately for participants with and without PPD.

Cortisol levels decreased significantly from the pre- to postnatal stage, but more so in participants without PPD. Participants with PPD had lower cortisol levels than those without. As shown in Figure 1, cortisol levels remained unchanged and low in participants with PPD from prenatal to postnatal stage (small effect size), while in participants without PPD, cortisol levels dropped from prenatal to postnatal stage (large effect size).

Cortisone levels also decreased significantly from pre- to postnatal stage, but more so in participants without PPD (though always large effect sizes within groups). Participants with PPD had lower cortisone levels that those without.

Testosterone levels neither changed significantly over time nor differed between pre- and postnatal stages (trivial or small effect sizes within groups).

Progesterone levels decreased in participants without PPD but increased in individuals with PPD (though always small effect sizes within groups). Compared to those with PPD, participants without PPD had higher progesterone levels.

DHEA levels decreased from the pre- to postnatal stages (always small effect sizes within groups). Participants with PPD had higher DHEA levels than participants without PPD.

### 3.3. Correlations between Symptoms of Depression and Steroid Levels (Prenatal and Postnatal Stage)

Table 3 reports the Pearson’s correlation coefficients between symptoms of depression (BDI, EPDS) and steroid levels (entire sample). Please see Table 3 for all the details. Here, we summarize the most important correlations.

Higher depression scores (BDI) were associated with lower prenatal cortisol, lower pre- and postnatal cortisone, lower progesterone (pre- and postnatal), and higher DHEA (pre- and postnatal).

Higher post-partum depression scores (EPDS) were associated with lower prenatal cortisol, lower cortisone (pre- and postnatal), lower progesterone (pre- and postnatal), and higher DHEA (pre- and postnatal).

Prenatal cortisol was associated with postnatal cortisol, and pre- and postnatal cortisone. Prenatal cortisol did not correlate with testosterone, progesterone, or DHEA.

Postnatal testosterone did not correlate with other steroids.

Higher prenatal progesterone was associated with higher postnatal progesterone.

Higher prenatal DHEA was associated with higher postnatal DHEA.

To summarize, more marked symptoms of depression were associated with lower cortisol, cortisone, and progesterone levels, both at the pre- and postnatal stages, and with higher DHEA levels, both at the pre- and postnatal stages. Cortisol levels were associated with cortisone levels. Testosterone was generally unrelated, except for higher prenatal progesterone levels.

### 3.4. Current Post-Partum Depression and Issues of Depression before Pregnancy

Participants with current PPD reported having had issues of depression before pregnancy (X^2^(*N* = 98, *df* = 1) = 24.52, *p* = 0.001). The odds of suffering currently from PPD was 38.11-fold higher in participants with issues of depression before pregnancy, compared to participants without issues of depression before pregnancy (OR = 38.11; CI (95%): 4.86–299.13).

### 3.5. Predicting Postnatal Depression Levels (Edinburgh Depression Rating Scale) Based on Prenatal Hair Steroids

A multiple regression analysis was performed to predict levels of postnatal depression (Edinburgh Depression Rating Scale) as a function of prenatal hair steroid levels. First, the Durban-Watson coefficient was satisfactory, and second, predictors explained 30% (*R*^2^ = 0.30) of the variance in postpartum depression.

As shown in Table 4, both lower prenatal cortisone and progesterone levels predicted postpartum depression levels, while prenatal cortisol, testosterone, and DHEA did not reach statistical significance.

### 3.6. Predicting Postnatal Depression Status on the Basis of Prenatal and Postnatal Hair Steroid Levels

A binary logistic regression analysis was performed to predict the postnatal depression status (yes vs. no) based on both prenatal and postnatal hair steroid levels. Compared to the default model, which correctly assigned 51% of participants as being depressed or not, the new model assigned 98% of participants correctly as being depressed or not. Predictors explained 92.4% of the variance in the dependent variable (Nagelkerke *R*^2^: 0.924). These predictors were as follows (as shown in Table 5): A participant was correctly assigned as postnatally depressed if prenatally lower levels of cortisol and progesterone, and higher levels of DHEA, were observed, and if postnatally lower levels of cortisol, cortisone, progesterone, and higher levels of DHEA were observed.

## 4. Discussion

The key findings of the present study were that, compared to women without post-partum depression (PPD), women with PPD had lower hair cortisol, cortisone, and progesterone levels and higher DHEA levels, both pre- and postnatally. Lower prenatal cortisone and progesterone levels predicted higher post-partum depression levels (continuous dimension). Prenatal low cortisol, cortisone, and progesterone, and higher DHEA levels, together with postnatal low cortisol, cortisone, and progesterone, and higher DHEA levels predicted status as participants with post-partum depression with an accuracy of 98%. Last, testosterone levels were unrelated to PPD. The present results add to the current literature in an important and new way, as specific combinations of hair steroids both pre- and postnatally were associated with PPD scores and PPD status 12 weeks after delivery.

Four hypotheses and two research questions were formulated, and each of these will now be considered.

Our first hypothesis was that, compared to healthy controls, women with PPD would release less cortisol, prenatally, and this was confirmed. The present findings, therefore, support the concept of hypocortisolemia, which appears to mirror a down-regulated (and, therefore, dysfunctional) HPA AA in women with PPD [11,21,22]. Importantly, this down-regulation was already present prenatally. In contrast, the present pattern of result is at odds with the studies that failed to find an association between lower cortisol levels and post-partum depression (see overview in [13,24,33]). The present findings are also at odds with the studies reporting increased hair cortisol concentrations in women with PPD [34]. While we accept that methodological issues (sample size, cortisol sampling, definition of PPD) might have led to previous contradictory findings, the present study was statistically robust and fulfilled all the quality criteria for a methodologically sound study (see Appendix A). Accordingly, the present results expand upon previous results in showing that hypocortisolemia was present at least 12 weeks before delivery in women with PPD.

Our second hypothesis was that, compared to the healthy controls, women with PPD would also release less cortisol postnatally, and this was also confirmed. The present findings, therefore, support the notion of hypocortisolemia.

Our third hypothesis was that levels of cortisol would remain unaltered and low in participants with PPD, while we expected a drop in cortisol levels from a prenatal high to postnatal low levels in healthy participants. Again, these expectations were fully confirmed (see Table 3 and Figure 1). The findings are, therefore, entirely in agreement with the notion of pregnancy as a transient and physiological period of hypercortisolism [69]. Among healthy pregnant women, cortisol secretion increases exponentially from the second to the third trimester [34] as a result of the placenta’s activity as an additional endocrine organ. Specifically, the placenta releases additional CRH into both the mother’s and fetus’s blood supply and triggers further cortisol release, which is not downregulated by the mineralocorticoid-receptor and glucocorticoid-receptor feedback loop [22,29]. At a behavioral level, a pregnant woman with a well-functioning stress response becomes less responsive to external stressors. In contrast, pregnant women with down-regulated HPA AA mirroring hypocortisolemia do not appear to be able to react adequately to external stressors, resulting in possible social withdrawal, reduced activity, and a depressed mood. Taken together, therefore, the present findings strongly suggest that it is inappropriate to assume that low cortisol secretion equates to positive psychological functioning and that high cortisol secretion indicates psychological issues [16].

Our fourth hypothesis was that participants currently suffering from PPD would also report that they had issues of depression in their past and before pregnancy. This prediction was confirmed. Participants with issues of depression before pregnancy had a 38.1-fold risk of suffering from PPD currently. Our findings support the proposition that PPD can be considered an additional expression of depression among women with general issues of depression [51,52,53,70]. In this respect, the present findings are in accord with those studies concluding that there are both biological and psychosocial predictors of PPD. Specifically, the strongest biological predictors are the dysregulation of the HPA AA, along with inflammatory processes and genetic vulnerabilities [12], while the strongest psychological predictors are severe life events, symptoms of depression before pregnancy [12], certain forms of chronic strain, low relationship quality, and low support from partner and mother [12,70].

Our first exploratory research question concerned whether other steroids (cortisone, testosterone, progesterone, DHEA) differed in level between the pre- and postnatal stages or between participants with and without PPD.

As reported in Table 3, testosterone differed neither between participants with and without PPD, nor from the pre- to the postnatal stage. Following Schiller et al. [48], testosterone concentrations have not been a focus of research looking for PPD-related biomarkers, and the results of the present study suggest that testosterone concentrations do not contribute to an endcrinological explanation of the occurrence of PPD.

Cortisone levels changed from the pre- to the post-natal stage, as did cortisol levels. While there is no direct evidence in the literature for such changes in cortisone concentrations in women with a higher risk of PPD, we assume that cortisol and cortisone concentrations go in parallel, as oxidation of cortisol leads to its inactive form, cortisone.

DHEA levels were higher in participants with PPD and declined over time. This result appears to be at odds with the observation that DHEA concentrations are lower among individuals with major depressive disorders [48]. However, Schiller et al. [48] also noted that associations between DHEA concentrations and the occurrence of PPD have not so far been shown. It follows that the present results appear to be the very first on the topic (and, above all, based on hair steroid analyses), and that further studies are needed to replicate the present findings.

For progesterone, this increased over time in participants with PPD, while it decreased in healthy participants. Again, to the best of our knowledge, this is the first study to investigate these hair steroids both pre- and postnatally and between participants, respectively, with and without PPD. Thus, the present findings might be used as a point of reference for future studies.

For the second exploratory research question, the combination of hair steroid levels both pre- and postnatally allowed us to predict with 98% accuracy a diagnosis of PPD. As shown in Table 4 and Table 5, in addition to pre- and postnatal low cortisol and low cortisone, pre- and postnatal low progesterone and high DHEA levels predicted the PPD status. It follows that the combination of several hair steroids both pre- and postnatally, rather than a single biomarker, allow accurate prediction of PPD 12 weeks after delivery.

Despite the novelty of these findings, the following limitations caution against their overgeneralization. First, in addition to a thorough clinical and psychiatric interview to ascertain the diagnosis of major depressive disorders, with the specifier of onset four weeks after delivery to establish the existence of post-partum depression, we employed the Edinburgh Postnatal Depression Scale, the most commonly used screening instrument to self-rate symptoms of PPD. The EPDS has some limitations. Its predictive power with respect to PPD is just 47%–64%, with a risk, therefore, of a substantial proportion of false positives [71]. However, to counter this, we used a cut-off score of 12 instead of 10 to assign participants to the PPD condition, and a score of 5 or lower for assignment to the healthy condition. As shown in Table 4 and Table 5, the wisdom of this choice of cut-off values was confirmed by hair steroid levels. Second, Hagen [72] noted that factors increasing the risk of PPD include, for example, poor quality of marital relations and demanding or challenging infant behavior. We did not check for such factors, and it is, therefore, possible that other unassessed and latent variables might have biased two or more dimensions in the same or opposite directions. Third, participants were asked if they had experienced symptoms of depression during their lifetimes and before pregnancy. Asking about symptoms of depression during the pregnancy might have provided further insights into the associations between the current psychological status during pregnancy and ante- and postnatal hair steroids concentrations. Fourth, from the perspective of evolutionary psychology, PPD is an adaptive strategy to save energy, to signal non-hostility, and to trigger receipt of resources from the social environment to help both needy mother and newborn. It would, therefore, have been useful to know whether or how the PPD status of the mother altered her relationship with her husband and other family members. Fifth, we assessed steroids in the hair, which, by definition, are not comparable to steroids in blood, urine, or saliva. We assessed steroids in hair using mass spectrometry, while in other studies blood, saliva, or urine cortisol concentrations were assessed with ELISA kits. It follows that comparisons of results between studies using different assessment tools with varying sensitivities and steroid concentrations, along with different samples and diagnostic criteria, might be problematic. Furthermore, as discussed extensively by Stalder et al. [19], hair steroid levels may vary as a function of season, sweating, and other confounding factors. Sixth, we assessed hair steroids of the last trimester and the first 12 weeks after delivery. Accordingly, the present results do not reflect steroid concentrations for all trimesters, and the results do not tell us more about hair steroid concentrations since the very beginning of pregnancy. However, this would have been particularly interesting. As reported in the present data, and as mentioned in elsewhere [51,52], women with post-partum depression already have a higher risk of reporting symptoms of depression before pregnancy. Accordingly, assessing hair steroids throughout pregnancy or even before pregnancy would have yielded a more comprehensive understanding of the underlying neuroendocrinological processes preceding PPD. Seventh, we assessed a small sample of women with and without PPD. Accordingly, the present results, particularly the exploratory ones, require replication with larger samples.

### Clinical Implications

The pattern of results suggests that the combination of symptoms of depression before pregnancy and low hair steroid concentrations at least 12 weeks before delivery could be employed to estimate the risk of the future mother suffering from symptoms of post-partum depression 12 weeks after delivery. Accordingly, pregnant women at risk might receive specific counseling and psychological support to prevent or to reduce the risk of PPD. This pattern of results further implies that we are highly unlikely to find one single biomarker to predict PPD. Rather, the combination of several hair steroids and the occurrence of symptoms of depression before pregnancy appears more promising.

## 5. Conclusions

The present data confirmed the notion that hypocortisolemia in women with post-partum depression lasts at least from the last 12 weeks before delivery to at least the following 12 weeks after delivery. Furthermore, pre- and postnatal hair steroid levels of cortisol, cortisone, progesterone, and DHEA, but not testosterone, showed different patterns in women with and without PPD. The combination of cortisol, cortisone, progesterone, and DHEA allowed precise prediction of a women’s status with respect to PPD.

## Figures and Tables

**Figure 1 jcm-08-01290-f001:**
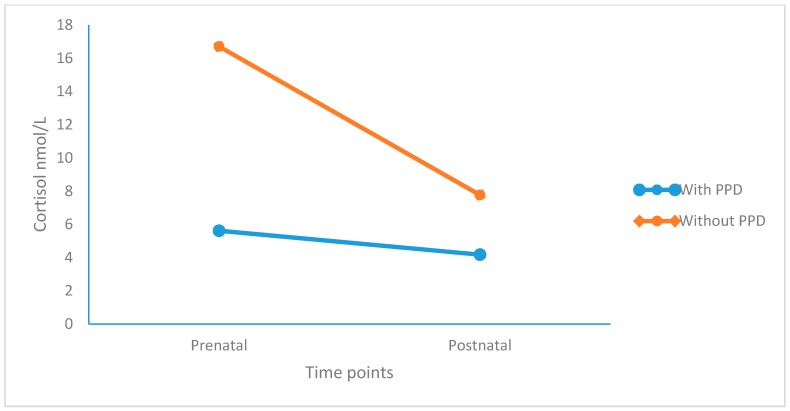
Cortisol levels decreased from prenatal to postnatal stage, but more so in participants without PPD than in those with PPD. In participants with PPD, cortisol secretion remained blunted from the prenatal to postnatal stage. Points are means.

**Table 1 jcm-08-01290-t001:** Descriptive and inferential statistical indices of hair steroids, separately for participants with (*n* = 48) and without (*n* = 50) postpartum depression.

				Group							Statistics	
	With PPD			Healthy Controls								
*N*	48	48		50	50							
Time period of steroid levels	Prenatal	Postnatal	*d*	Prenatal	Postnatal	*d*						
Hair steroids	M (SD)	M (SD)		M (SD)	M (SD)		Time		Group		Time × Group interaction	
							F	*η* _p_ ^2^	F	*η* _p_ ^2^	F	*η* _p_ ^2^
Cortisol	5.62 (3.98)	4.18 (2.59)	0.43 (S)	16.71 (9.51)	7.77 (5.47)	1.15 (L)	40.43 ***	0.296 (L)	10.47 ***	0.373 (M)	21.08 ***	0.180 (L)
Cortisone	12.48 (7.57)	6.14 (3.50)	1.08 (L)	48.49 (26.35)	17.94 (12.83)	1.47 (L)	59.44 ***	0.282 (L)	17.97 ***	0.158 (L)	25.63 ***	0.211 (L)
Testosterone	0.34 (0.53)	0.42 (0.45)	0.16 (T)	0.58 (0.79)	0.82 (0.95)	0.27 (S)	1.00	0.010 (S)	1.52	0.016 (S)	0.79	0.008 (S)
Progesterone	30.66 (26.44)	40.58 (25.37)	0.38 (S)	107.59 (45.03)	86.91 (51.55)	0.42 (S)	0.60	0.006 (S)	19.13 ***	0.166 (L)	4.80 *	0.049 (S)
DHEA	7.74 (4.56)	6.34 (5.37)	0.28 (S)	3.93 (2.97)	2.95 (2.64)	0.34 (S)	3.53 ^(^*^)^	0.035 (S)	7.48 **	0.072 (M)	0.11	0.001 (S)

Notes: Values: Always nmol/L. PPD = postpartum depression; DHEA = dehydroepiandrosterone ^(^*^)^ = *p* < 0.10; * = *p* < 0.05; ** = *p* < 0.01; *** = *p* < 0.001; (T) = trivial effect size; (S) = small effect size, (M) = medium effect size; (L) = large effect size. Statistics always controlling for baseline values.

**Table 2 jcm-08-01290-t002:** Descriptive and inferential statistical indices of sociodemographic, depression- and delivery-related information, separately for participants with post-partum depression (PPD) (*N* = 48), and without post-partum depression (*N* = 50).

	Groups		Statistics	
	Individuals with PPD	Individuals without PPD		
*N*	48	50		
	*n* (%)	*n* (%)	*X*^2^-test	
Depressive episode in life (yes/no)	21/27	1/49	*X*^2^ (*N* = 98, *df* = 1) = 24.52 ***	
	M (SD)	M (SD)	*t*-tests	Cohen’s *d*
Age (years)	25.88 (4.28)	25.22 (4.88)	*t*(96) = 0.71	0.14 (S)
Beck Depression Inventory	28.08 (9.19)	4.30 (3.13)	*t*(96) = 17.28 ***	3.49 (L)
Edinburgh Postnatal Depression Scale	15.69 (3.47)	3.82 (2.76)	*t*(96) = 18.77 ***	3.78 (L)
Gestational age (days)	270.23 (12.79)	274.04 (11.92)	*t*(96) = 1.53	0.00 (-)
Apgar score at delivery	8.81 (0.70)	9.98 (0.25)	*t*(96) = 1.58	2.23 (L)
Apgar score 10′ after delivery	9.85 (0.50)	9.96 (0.20)	*t*(96) = 1.38	0.29 (S)

Notes *** = *p* < 0.001. (S) = small effect: t size; (M) = medium effect size; (L) = large effect size.

**Table 3 jcm-08-01290-t003:** Correlations between symptoms of depression and hair steroids (*N* = 98).

				Dimensions									
		1	2	3	4	5	6	7	8	9	10	11	12
1	BDI	-	0.90 ***	−0.28 **	−0.13	−0.36 ***	−0.23 *	−0.14	−0.08	−0.40 ***	−0.23 *	0.21 *	0.21 *
2	EPDS		-	−0.23 **	−0.15	−0.35 **	−0.24 *	−0.16	−0.14	−0.40 ***	−0.22 *	0.19 ^(^*^)^	0.20 *
3	Cortisol pre			-	0.91 ***	0.94 ***	0.86 ***	−0.12	−0.04	0.04	−0.02	0.07	0.11
4	Cortisol post				-	0.86 ***	0.04 ***	−0.09	−0.03	−0.01	−0.03	0.07	0.17
5	Cortisone pre					-	0.91 ***	−0.15	−0.07	0.03	−0.03	0.00	0.09
6	Cortisone post						-	−0.11	−0.08	−0.02	−0.03	0.00	0.11
7	Testosterone pre							-	0.90 ***	0.61 ***	0.22 *	0.01	0.00
8	Testosterone post								-	0.05	−0.05	−0.04	−0.02
9	Progesterone pre									-	0.65 ***	−0.16	−0.14
10	Progesterone post										-	−0.15	−0.10
11	DHEA pre											-	0.65 **
12	DHEA post												-

Notes: BDI = Beck Depression Inventory; EPDS = Edinburgh Postpartum Depression Scale; DHEA = dehydroepiandrosterone; ^(^*^)^ = *p* < 0.10; * = *p* < 0.05; ** = *p* < 0.01; *** = *p* < 0.001.

**Table 4 jcm-08-01290-t004:** Multiple linear regression with post-partum depression scores as dependent variables (Edinburgh Postnatal Depression Scale) and antenatal hair steroid levels (cortisol; cortisone; testosterone; progesterone; DHEA) as predictors.

Dimension	Variables	Coefficient	Standard Error	Coefficient β	*t*	*p*	*R*	*R* ^2^	Durbin-Watson Coefficient
Edinburgh Postnatal Depression Scale	Intercept	12.69	1.038	-	12.22	0.001	0.552	0.30	1.78
	Antenatal cortisol	0.152	0.111	0.127	1.373	0.173			
	Antenatal cortisone	−0.105	0.041	−0.664	−2.548	0.012			
	Antenatal testosterone	0.219	1.112	0.022	0.197	0.884			
	Antenatal progesterone	−0.032	0.009	−0.393	−3.453	0.001			
	Antenatal DHEA	0.099	0.085	0.1.06	1.164	0.248			

Notes: DHEA = dehydroepiandrosterone.

**Table 5 jcm-08-01290-t005:** Binary logistic regression with post-partum depression (yes vs. no) as dependent variable, and antenatal hair steroid levels (cortisol; cortisone; testosterone; progesterone; DHEA) as predictors.

Dimension	Variables	Coefficient	Standard Error	Wald	*p*	Nagelkerke *R*^2^
Depression (yes vs. no)	Intercept	2.99	1.074	7.729	0.003	0.924
	Cortisol prenatal	−0.526	0.150	12.366	0.000	
	Cortisol postnatal	1.083	0.307	12.471	0.001	
	Cortisone postnatal	−0.304	0.131	5.440	0.020	
	Progesterone prenatal	−0.019	0.007	7.043	0.008	
	Progesterone postnatal	−0.024	0.011	4.759	0.029	
	DHEA prenatal	0.211	0.099	4.525	0.033	
Excluded variables	Cortisone prenatal, testosterone prenatal, testosterone postnatal, DHEA postnatal, (all Wald’s < 1.8, all *p*’s > 0.10).					

Notes: DHEA = dehydroepiandrosterone.

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
