# Peer review of "Prenatal and Postnatal Hair Steroid Levels Predict Post-Partum Depression 12 Weeks after Delivery"

_jcm, 2019, doi:10.3390/jcm8091290_

Round 1
Reviewer 1 Report
The authors have done a very good job and the manuscript have been improved. I recommend to check the spelling of some words.
Author Response
Dear Reviewer,
We thank Reviewer 1 for their valuable comments; we are pleased to know that we did a very good job.Per request, we have checked the spelling of the entire text once again.
Reviewer 2 Report
Although the authors have revised the manuscript and provided additional details for the methods section, several issues remain; the writing is still unclear in many parts, it is difficult to interpret many of the primary findings, and the revisions brought to light additional concerns.
Specifically:
The introduction is still not cohesive; the authors introduce the purpose of their study regarding hair cortisol sample, but then the next headings are about cortisol levels through blood, saliva and urine without any introduction to the methods used for cortisol sampling. Throughout each section (particularly the cortisol and other steroid sections), there is no real summary of results; just a listing of what prior studies had found (many of which are contradictory). It does not flow well or succinctly summarize findings for the reader. I disagree with the justification and use of stepwise multiple regression. Please refer to the citations below regarding the numerous limitations and shortcomings of using this technique. Your models should be based on theory; not guided by statistics themselves. Further, the limitations of using this analysis have not been acknowledged. Smith, G. (2018). Step away from stepwise. Journal of Big Data, 5(1), 32. Mundry, R., & Nunn, C. L. (2008). Stepwise model fitting and statistical inference: turning noise into signal pollution. The American Naturalist, 173(1), 119-123. Whittingham, M. J., Stephens, P. A., Bradbury, R. B., & Freckleton, R. P. (2006). Why do we still use stepwise modelling in ecology and behaviour? Journal of animal ecology, 75(5), 1182-1189. The present results are limited to the final trimester; given that the introduction focuses on timing of assessment (between first and second trimesters, etc.), the implications of steroid assessment only in the final trimester of this study should also be discussed. The inclusion criteria for PPD require that subjects report 12 or higher on the EPDS; and the inclusion criteria for no PPD require an EPDS score of 5 or lower. Similar gaps in criteria are present for the BDI; yet no one seemed to fall into the gap categories given that 65 women were categorized as PPD and 389 were categorized without PPD. It seems strange that no one would have categorized by EPDS of 5-12. Is that correct? Please provide a justification for and additional information about the random selection of 50 women without PPD- why was the full sample not used? Was there a program that was used to assist with the random selection? The large sample size estimate stems from an article that used EPDS scores within 96 hours of delivery and again at an outpatient appointment. This effect size does not apply to hair steroid levels. Please justify. Please cite the cutoff score of 12 on the EPDS. The validated score appears to be either 10 or 13. Matthey, S., Henshaw, C., Elliott, S., & Barnett, B. (2006). Variability in use of cut-off scores and formats on the Edinburgh Postnatal Depression Scale–implications for clinical and research practice. Archives of women's mental health, 9(6), 309-315. Expand the limitations section regarding the sampling technique (e.g., the group without PPD does not represent the full sample); further, the sample size is relatively small, and these results (particularly the exploratory ones) require replication with larger sample
Minor revisions include:
As currently written, the abstract does not sufficiently explain the procedure; this was a cross-sectional study conducted 12-weeks after delivery. At this time, subjects gave hair sample (reflecting 12 weeks pre-and post-delivery) and completed surveys. Sleep is still referred to in section 2.8 & table 3 Some information from section 2.6 should be moved to the Introduction section (e.g., the justification for using hair sampling).
Author Response
Dear Reviewer,
Thank you for reviewing the manuscript. Your kind efforts were highly appreciated.
Reviewer 2 |
|
Although the authors have revised the manuscript and provided additional details for the methods section, several issues remain; the writing is still unclear in many parts, it is difficult to interpret many of the primary findings, and the revisions brought to light additional concerns. |
We thank the Reviewer 2 for their scrutiny and valuable comments. We believe that her/his comments helped us again to make significant improvement to the quality of the manuscript. |
Specifically: |
|
The introduction is still not cohesive; the authors introduce the purpose of their study regarding hair cortisol sample, but then the next headings are about cortisol levels through blood, saliva and urine without any introduction to the methods used for cortisol sampling. Throughout each section (particularly the cortisol and other steroid sections), there is no real summary of results; just a listing of what prior studies had found (many of which are contradictory). It does not flow well or succinctly summarize findings for the reader. |
Per request, we introduced a short opening paragraph: In the following sections, we provide a brief overview of studies on steroids during and after pregnancy. Such steroids, and above all, cortisol, were mainly assessed in blood, saliva and urine, and to a much lesser extent in hair.
Subheading: Plasma cortisol in healthy pregnant women Summary: Thus, increases in plasma cortisol from the second to the third trimester reflect physiological adaptations of pregnancy among healthy women.
Subheading: Salivary, blood and urine cortisol in pregnant women with major depressive disorders Summary: Collectively, the pattern of results from cortisol concentrations in saliva, blood and urine in pregnant women with PPD does not seem to be fully consistent. Nevertheless, a hypercortisolemia was mainly associated with transient symptoms of depression, while a hypocortisolemia was associated with chronic symptoms of depression.
Subheading: Hair cortisol in women with post-partum depression Summary: To summarize, studies on hair cortisol during and after pregnancy are scarce and the pattern of results appears inconsistent. Such inconsistent findings appear to reflect above all methodological issues such as the timing of psychological assessments, and the lack of a thorough diagnosis of PPD.
Subheading: Other steroids during pregnancy and after delivery Summary: To conclude, research on further steroids such a cortisone, progesterone, testosterone or DHEA is scarce, and results are inconsistent and based on blood and saliva samples, while data from hair samples are missing so far. |
I disagree with the justification and use of stepwise multiple regression. Please refer to the citations below regarding the numerous limitations and shortcomings of using this technique. Your models should be based on theory; not guided by statistics themselves. Further, the limitations of using this analysis have not been acknowledged. Smith, G. (2018). Step away from stepwise. Journal of Big Data, 5(1), 32. Mundry, R., & Nunn, C. L. (2008). Stepwise model fitting and statistical inference: turning noise into signal pollution. The American Naturalist, 173(1), 119-123. Whittingham, M. J., Stephens, P. A., Bradbury, R. B., & Freckleton, R. P. (2006). Why do we still use stepwise modelling in ecology and behaviour? Journal of animal ecology, 75(5), 1182-1189. |
We thank Reviewer 2 for drawing out attention to the statistical issues related to stepwise multiple regression analyses. We admit that we have been unaware of the literature raising concerns as regards the method of a stepwise multiple regression analysis. We have learned from Harrell (2001) that stepwise multiple regression analyses bear the risk that standard errors are biased toward 0, that p-values are biased toward 0, and that parameters estimates are biased away from 0. We have performed a ‘simple’ multiple regression analysis. We have modified and adapted Table 4: we have reported the full result model, including also those variables, which did not reach statistical significance. The overall pattern of results did not change. |
The present results are limited to the final trimester; given that the introduction focuses on timing of assessment (between first and second trimesters, etc.), the implications of steroid assessment only in the final trimester of this study should also be discussed |
We thank Reviewer 2 for this candid comment. In the Limitation section, the text reads as follows: Sixth, we assessed hair steroids of the last trimester and the first 12 weeks after delivery; accordingly, the present results do not reflect steroid concentrations of the all trimesters, and the results do not tell us more about hair steroid concentrations since the very beginning of pregnancy. However, this would have been particularly interesting: as reported in the present data, and as mentioned in xxx, women with post-partum depression have a higher risk to report symptoms of depression already before pregnancy. Accordingly, assessing hair steroids throughout pregnancy or even before pregnancy would have allowed to get a more comprehensive understanding of the underlying neuroendocrinological processes preceding a PPD. |
Please cite the cutoff score of 12 on the EPDS. The validated score appears to be either 10 or 13. Matthey, S., Henshaw, C., Elliott, S., & Barnett, B. (2006). Variability in use of cut-off scores and formats on the Edinburgh Postnatal Depression Scale–implications for clinical and research practice. Archives of women's mental health, 9(6), 309- 315. |
We mentioned that the cut-off score of 12 points was used. The text reads now: However, we employed the more restrictive cut-off score of 12 for the self-rating of post-partum depression; others recommend to use cut-off score of 13 {Matthey, 2006 #4358}. |
Expand the limitations section regarding the sampling technique (e.g., the group without PPD does not represent the full sample); further, the sample size is relatively small, and these results (particularly the exploratory ones) require replication with larger sample |
In the Limitation section, the text reads as follows: Seventh, we assessed a smaller sample of women with and without PPD; accordingly, the present results, particularly the exploratory ones, require replication with larger samples. |
The inclusion criteria for PPD require that subjects report 12 or higher on the EPDS; and the inclusion criteria for no PPD require an EPDS score of 5 or lower. Similar gaps in criteria are present for the BDI; yet no one seemed to fall into the gap categories given that 65 women were categorized as PPD and 389 were categorized without PPD. It seems strange that no one would have categorized by EPDS of 5-12. Is that correct? Please provide a justification for and additional information about the random selection of 50 women without PPD- why was the full sample not used? |
Thank you! We specified this point, and the text reads now: Additionally, of the 495 women approached, 41 (8.3%) did neither fulfill the criteria for participants with or without PPD, and 389…” The following information were added: Randomization occurred based on the following criteria: matching the group of participants with PPD as regards sample size, age, gestational age, and Apgar-score 10’ after delivery. A total of 167 participants fulfilled these criteria; their codes were put in sealed and separate envelopes; sealed envelopes were put in an opaque ballot box and stirred. A staff member of the hospital not otherwise involved in the study picked out 50 envelopes. To avoid possible and further confounders, we tested the sample of 48 participants with PPD against 50 participants without PPD. |
Was there a program that was used to assist with the random selection? The large sample size estimate stems from an article that used EPDS scores within 96 hours of delivery and again at an outpatient appointment. This effect size does not apply to hair steroid levels. Please justify. |
Thank you for drawing our attention to this point. To avoid that the sample was statistically underpowered, we have chosen a study with higher effect sizes. Further, neighter Caparros-Gonzalez et al, nor Braig et al, nor Orta et al., which assessed hair steroids assessed women with a firmly performed diagnose of post-partum depression. |
Minor revisions include: |
|
As currently written, the abstract does not sufficiently explain the procedure; this was a cross-sectional study conducted 12-weeks after delivery. At this time, subjects gave hair sample (reflecting 12 weeks pre-and post-delivery) and completed surveys. |
We have adapted the abstract, which reads as follows: delivery and with or without PPD. Method. The present study was a cross-sectional study conducted twelve weeks after delivery. At that time, 48 women (mean age: 25.9 years) with PPD and 50 healthy controls (mean age: 25.2 years) completed questionnaires on depressive symptoms. Further, at the same time point, 6cm lengths of hair strands were taken, providing samples of hair steroids 12 weeks before and 12 weeks after delivery in order to analyze hair steroids (cortisol, cortisone, progesterone, testosterone, DHEA). |
Sleep is still referred to in section 2.8 & table 3 |
Thank you! We have carefully checked the text once again, and we have deleted all information as regards sleep. |
Some information from section 2.6 should be moved to the Introduction section (e.g., the justification for using hair sampling). |
As per request, we have moved the first paragraph of section 2.6. Hair strands sampling to the Introduction section. This was an excellent suggestion of Reviewer 2, as it further helped us to justify the methodological approach used in the present study. |
Harrell, F. E. (2001), Regression modeling strategies: With applications to linear models, logistic regression, and survival analysis, Springer-Verlag, New York.

Round 2
Reviewer 2 Report
No further comments